# Assessing the Association between Biomarkers and COVID-19 Mortality Using the Joint Modelling Approach

**DOI:** 10.3390/life14030343

**Published:** 2024-03-06

**Authors:** Matteo Di Maso, Serena Delbue, Maurizio Sampietro, Monica Ferraroni, Annalisa Modenese, Maria Dolci, Federico Ambrogi, Pasquale Ferrante

**Affiliations:** 1Department of Clinical Sciences and Community Health, Branch of Medical Statistics, Biometry and Epidemiology “G.A. Maccacaro”, Università degli Studi di Milano, 20133 Milan, Italy; matteo.dimaso@unimi.it (M.D.M.); monica.ferraroni@unimi.it (M.F.); 2Department of Biochemical, Surgical & Dental Sciences, Università degli Studi di Milano, 20122 Milan, Italy; serena.delbue@unimi.it (S.D.); maria.dolci@unimi.it (M.D.); pasquale.ferrante@unimi.it (P.F.); 3Istituto Clinico Città Studi, 20131 Milan, Italy; maurizio.sampietro@ic-cittastudi.it (M.S.); annalisa.modenese@ic-cittastudi.it (A.M.); 4Fondazione IRCCS Ca’ Granda, Ospedale Maggiore Policlinico, 20122 Milan, Italy; 5Laboratory of Biostatistics and Data Management, IRCCS Policlinico San Donato, San Donato Milanese, 20097 Milan, Italy

**Keywords:** COVID-19 mortality, biomarkers, time-varying covariates, joint modelling approach, survival analysis

## Abstract

We evaluated the association between biomarkers and COVID-19 mortality. Baseline characteristics of 403 COVID-19 patients included sex and age; biomarkers, measured throughout the follow-up, included lymphocytes, neutrophils, ferritin, C-reactive protein, glucose, and LDH. Hazard ratios (HRs) and corresponding 95% credible intervals (CIs) were estimated through joint models (JMs) using a Bayesian approach. We fitted univariable (a single biomarker) and multivariable (all biomarkers) JMs. In univariable analyses, all biomarkers were significantly associated with COVID-19 mortality. In multivariable analysis, HRs were 1.78 (95% CI: 1.13–2.87) with a doubling of neutrophils levels, 1.49 (95% CI: 1.19–1.95) with a doubling of C-reactive protein levels, 2.66 (95% CI: 1.45–4.95) for an increase of 100 mg/dL of glucose, and 1.31 (95% CI: 1.12–1.55) for an increase of 100 U/L of LDH. No evidence of association was observed for lymphocytes and ferritin in multivariable analysis. Men had a higher COVID-19 mortality risk than women (HR = 1.75; 95% CI: 1.07–2.80) and age showed the strongest effect with a rapid increase from 60 years. These findings using JM confirm the usefulness of biomarkers in assessing COVID-19 severity and mortality. Monitoring trend patterns of such biomarkers can provide additional help in tailoring the appropriate care pathway.

## 1. Introduction

The coronavirus disease 2019 (COVID-19) pandemic, caused by severe acute respiratory syndrome coronavirus 2 (SARS-CoV-2), emerged in late 2019, and since then, has been threatening human health. As of 17 December 2023, COVID-19 has affected more than 772 million people and globally, nearly 7 million died [1]. Italy was the first European country to be hit by the COVID-19 pandemic outbreak, with clusters of cases detected and the first COVID-19-attributed deaths in Lombardy.

Over time, evidence has shown that older age is the main predictor of COVID-19 severity and mortality [2,3]. In addition, a meta-analysis comprising more than 36,000 patients reported that men experienced the disease more severely (approximately 20%) and were also at a higher risk of death from COVID-19 (nearly 50%) than women [4]. The presence of pre-existing comorbidities, such as obesity [5,6,7,8,9], cardiovascular diseases [10,11,12,13], hypertension [11,13,14,15,16], diabetes [13,17,18,19,20], chronic obstructive pulmonary diseases [11,12,14,21], and cancer [12,14,22], are more common in patients with severe COVID-19, impacting their survival [23]. Limited evidence has also suggested the role of pre-existing chronic kidney diseases [24] and cerebrovascular diseases [11] in developing severe stages of COVID-19. Other recognized predictors for COVID-19 severity and mortality include hematological, coagulation, hematic, and biochemical markers [25,26,27]. The most investigated were lymphocytes, neutrophils, eosinophils (hematological), D-dimer, prothrombin time, activated partial-thromboplastin time (coagulation), aspartate aminotransferase, alanine aminotransferase (hematic), ferritin, C-reactive protein, procalcitonin, and lactate dehydrogenase (biochemical).

During the first wave of the COVID-19 pandemic, physicians at the Istituto Clinico Città Studi in Milan collected a set of biomarkers over a follow-up period, to understand whether changes in their levels might be used as prognostic factors for severity and mortality of COVID-19.

Thus, we evaluated here the association between a set of biomarkers measured throughout the follow-up and COVID-19 mortality using the joint modelling (JM) approach, the candidate tool for this kind of data.

## 2. Materials and Methods

### 2.1. Study Design and Patients

Between February and May 2020, a total of 403 COVID-19 patients were admitted to the Istituto Clinico Città Studi in Milan. Patients aged 21–100 years (mean of 72.4 and standard deviation of 17.0 years) and 58.3% were men. Baseline characteristics included the sex and age of patients, whereas the set of biomarkers included hematological, coagulation, and biochemical markers. Hematological markers included lymphocytes (count × 10^3^/µL) and neutrophils (count × 10^3^/µL); coagulation markers included D-dimer (ng/mL); and biochemical markers included ferritin (ng/mL), C-reactive protein (mg/L), glucose (mg/dL), and lactate dehydrogenase—LDH (U/L). The biomarker data were extracted from the patient’s clinical charts and stored in a database.

Person-time at risk (expressed in days) was computed as the time elapsed from the day of hospital admission to the day of COVID-19 death (event time), either to the day of hospital discharge or to the day of moving to other health care facilities (right-censoring time), whichever came first.

### 2.2. Statistical Analyses

We evaluated the association between biomarkers and COVID-19 mortality using JM proposed by Rizopoulos [28]. Particularly in the first epidemic outbreak, physicians did not have standard clinical protocols for the management of COVID-19 patients and for this reason, measurements of biomarkers were highly incomplete, especially at the baseline. In this context, the classical time-to-event analysis using Cox regression [29,30] with time-invariant covariates (i.e., variables that do not change value during the follow-up) is unfeasible, whereas the time-dependent Cox model [31] would likely lead to biased estimates in assessing the association between such incomplete time-varying covariates (i.e., variables that change value during the follow-up) and mortality risk.

The JM is suitable for estimating the association between time-varying covariates (i.e., biomarkers) and a time-to-event outcome (i.e., death from COVID-19). Briefly, the JM consists of two sub-models: (i) the survival sub-model used to estimate hazards for a set of time-invariant covariates (i.e., baseline characteristics) and (ii) the longitudinal sub-model used to predict the complete trajectories of time-varying covariates (i.e., biomarkers) during the follow-up, possibly considering a set of time-invariant covariates. The two sub-models are interdependent by means of a set of random effects, also called shared parameters. The shared parameters (i.e., patient-specific predicted trajectories of time-varying covariates) derive from the longitudinal sub-model and they are plugged into the survival sub-model. We used a Bayesian approach for fitting JMs. In particular, the estimation of JM’s parameters proceeded using the Markov chain Monte Carlo (MCMC) algorithm. The posterior distribution of the model parameters is derived under the assumption that given the shared parameters, both longitudinal and survival sub-models are assumed independent, and the longitudinal outcomes of each patient are assumed independent. For explorative purposes, we set independent and non-informative priors for baseline characteristics and shared parameters.

We fitted univariable and multivariable JMs. Univariable JMs included time-invariant covariates (i.e., age and sex) and a single time-varying covariate (i.e., a biomarker), whereas the multivariable JM included all covariates (i.e., all time-invariant and time-varying covariates). Summary statistics (i.e., the posterior mean and the credible interval of the posterior mean) were estimated for each parameter. In particular, the 95% equal-tailed credible intervals (CIs) were computed. In addition, to test the significance (at 5%) of parameter estimates, we used the 2 times probability that a parameter estimate is strictly positive or negative, whichever is less probable (*P*). The logarithmic transformation was used to account for the skewness in the distribution of lymphocytes (log-lymphocytes), neutrophils (log-neutrophils), D-dimer (log-D-dimer), ferritin (log-ferritin), and C-reactive protein (log-C-reactive protein). In addition, glucose and LDH levels were rescaled by dividing by 100 (glucose/100; LDH/100) for numerical stability. To account for possible non-linear effects, we modelled age and patient-specific biomarkers trajectories through the follow-up time by means of natural cubic splines (ns) with two knots. Two knots are generally sufficient to detect mild non-linear relationships and to avoid over-parametrization of the model considering the available sample size. Due to the high number of patients (78; 19%) with missing values for D-dimer during the whole follow-up period, we excluded it from the multivariable analysis. Analyses were performed using the JMbayes package [32] in R Statistical Software, version 4.0.5 (R Core Team 2021, Vienna, Austria).

## 3. Results

Among 403 COVID-19 patients admitted to the Istituto Clinico Città Studi, 140 died during the follow-up. Among the 263 patients who survived, 99 were discharged and 164 were moved to other health care facilities. The median follow-up was 14 days (range: 0–78 days).

Table 1 reports the distribution of patients, hazard ratios (HRs), and corresponding 95% confidence intervals (CIs) of COVID-19 mortality according to age group separately for men and women. HRs and corresponding 95% CIs were estimated using time-invariant Cox regression models including only age (in categories) and considering <60 years as reference. A higher mortality was observed for elderly patients compared with patients aged < 60 years, with a similar trend for men and women. Among men, the HRs run from 3.55 (95% CI: 1.07, 11.81) for patients aged 60–69 years to 26.88 (95% CI: 8.52, 84.81) for patients aged 90 years or older; among women, the corresponding figures were 1.84 (95% CI: 0.11, 29.41) and 18.74 (95% CI: 2.51, 139.91).

Table 2 reports estimates of univariable and multivariable JMs. Regarding the longitudinal sub-model of univariable JMs, the predicted log-lymphocyte count estimates significantly increased through the follow-up time (Figure 1, first panel) according to the positive natural cubic spline (ns) coefficients of time at measurement (Table 2). Conversely, the predicted log-lymphocyte count estimates significantly decreased with increasing age (according to the negative natural cubic spline coefficients for age reported in Table 2) and men showed lower predicted log-lymphocyte count estimates than women (according to the negative coefficient for sex in Table 2), though not significant.

Instead, the predicted log-neutrophils count estimates slightly decreased during the follow-up time (Figure 1, second panel), they increased with age and men showed higher predicted levels than women (Table 2). Likewise, the predicted log-D-dimer and log-ferritin estimates decreased through time (Figure 1, third and fourth panels), increased with age and men had higher levels (Table 2). For log-C-reactive protein, predicted levels showed a mixed trend through time (Figure 1, fifth panel). In particular, levels initially decreased according to the negative natural cubic spline coefficient of time at measurement concerning the first part of follow-up and increased thereafter (Table 2). The predicted log-C-reactive protein levels increased with age and men showed higher levels than women. Predicted levels of glucose and LDH significantly decreased during the follow-up (Figure 1, sixth and seventh panels). Only for LDH, predicted levels significantly increased with age and men had higher levels (Table 2).

Regarding the survival submodel of univariable JMs reported in Table 2, all biomarkers were significantly associated with COVID-19 mortality. An increase in the levels of biomarkers was associated with an increase in the mortality risk, except for lymphocytes (according to the negative log-hazard ratio in Table 2). In particular, the doubling of lymphocyte count levels was associated with approximately halving mortality risk with an HR of 0.58 (95% CI: 0.46–0.73). This HR was obtained by exp(−0.78 × 0.693), which is the exponential product of the log-hazard ratio estimate (−0.78) and the logarithm of 2 (0.693) corresponding to the doubling of lymphocyte count levels in the original scale. The strongest associations were observed for neutrophils (HR = 2.95; 95% CI: 2.31–3.76 for doubling of levels), C-reactive protein (HR = 2.53; 95% CI: 2.07–3.18 for doubling of levels), and glucose (HR = 3.16; 95% CI: 2.18–4.53 for an increase of 100 mg/dl). The HR for glucose was obtained by exp(1.15).

In the multivariable JM, there was no more evidence of association with COVID-19 mortality for ferritin and lymphocytes (survival sub-model of the multivariable JM in Table 2). Furthermore, the strength of the association was attenuated for the other biomarkers considered. In particular, the HRs were 1.78 (95% CI: 1.13–2.87) for doubling of neutrophils levels, 1.49 (95% CI: 1.19–1.95) for doubling of C-reactive protein levels, 2.66 (95% CI: 1.45–4.95) for an increase of 100 mg/dL in glucose levels, and 1.31 (95% CI: 1.12–1.55) for an increase of 100 U/L in LDH. Lastly, men had a nearly 2-fold higher risk than women (HR = 1.75; 95% CI: 1.07–2.80) and age showed the strongest effect on COVID-19 mortality with HRs starting to rapidly increase approximately from 60 years for both men and women (Figure 1, last panel).

## 4. Discussion

We used JM to evaluate the association between a set of biomarkers and COVID-19 mortality including some baseline patients’ characteristics. Patients were admitted to a hospital in Milan during the first wave of the COVID-19 pandemic outbreak. In the multivariable JM, increasing levels of some of the investigated biomarkers (i.e., neutrophils, C-reactive protein, glucose, and LDH) were significantly associated with higher mortality. In addition, men were at a higher risk of dying than women and the strongest association was observed for increasing age.

Previous findings on biomarkers showed their association with the severity and mortality of COVID-19 [2,3,25,27]. There is wide evidence of lymphopenia among COVID-19 patients [17,33,34,35,36,37,38,39]. Furthermore, some studies also reported a lower absolute number of lymphocytes in patients with more severe illness, compared with patients with mild illness [37,40,41]. Accordingly, we found a lower mortality risk for increasing the number of lymphocytes. Higher neutrophil count at hospital admission was reported in patients with a severe or critical disease stage compared with patients with a mild or moderate stage of COVID-19 [35,38]. Progressive increases in the number of neutrophils were associated with death [38]. Our predicted means of neutrophil count for patients who died followed a similar trend during the follow-up. In particular, the predicted means were 8.02 (95% CI: 6.59, 9.76) at baseline, 9.39 (95% CI: 7.72, 11.43) at day 7, and 10.07 (95% CI: 8.27, 12.25) at day 15 of follow-up; no further increase in predicted means was observed for subsequent follow-up times. Elevated D-dimer levels were frequently reported in COVID-19 patients [40,42,43,44]. Several meta-analyses showed the prognostic value of D-dimer for disease severity and mortality [1,45,46,47,48,49,50,51,52,53,54]. In addition, two studies reported that a baseline D-dimer level of >2 µg/mL was associated with a higher mortality [55,56]. Among patients who died in the present study, the predicted mean of D-dimer at day 7 of follow-up was 1.75 µg/mL and increasing the levels resulted in a higher mortality risk. Elevated levels of ferritin were associated with the progression to severe stages of COVID-19, as well as mortality [54,57,58,59]. A meta-analysis investigated the prognostic value of different biomarkers of anaemia and iron metabolism (including ferritin) in COVID-19 patients [60]. Based on the findings of 18 observational studies, comprising more than 7000 patients, the authors showed approximately a 2-fold higher pooled mean of ferritin levels in non-survivors (1303.08 ng/mL; 95% CI: 1072.26, 1533.90 ng/mL) than survivors (650.67; 95% CI: 541.84, 759.51 ng/mL). Likewise, our predicted mean levels of ferritin at day 7 of follow-up were nearly 2 times higher in patients who died (1058.22; 95% CI: 740.25, 1512.77 ng/mL) than in patients who survived (556.90; 95% CI: 481.32, 644.35 ng/mL). In the multivariable JM, however, there was no evidence of an association between ferritin and the risk of COVID-19 mortality. High levels of C-reactive protein were associated with the development of severe COVID-19 stages and higher mortality [1,46,49,51,53,54,57]. A C-reactive protein level of >10 mg/L has been shown to be a predictor of poor outcome [54]. In our analysis, the predicted mean levels of C-reactive protein were 10.7 mg/L (95% CI: 6.60, 17.38 mg/L) for non-survivors and 2.22 mg/L (95% CI: 1.66, 2.98 mg/L) for survivors. Furthermore, a meta-analysis showed a 4-fold higher risk of severe disease for levels of C-reactive protein >10 mg/L [61]. Accordingly, we estimated a mortality risk of approximately 2.5 times (univariable JM) and 1.5 times (multivariable JM) higher for the doubling of C-reactive protein levels. Higher levels of glucose were observed in the severe and critical groups of COVID-19 patients [25,51]. Among the biomarkers considered in the present analysis, glucose showed the strongest association with COVID-19 mortality considering the effect of other biomarkers (HR from multivariable JM = 2.66; 95% CI: 1.45–4.95 for an increase of 100 mg/dL). Lastly, elevated LDH levels have been reported in COVID-19 patients with the highest levels for patients with severe disease [1,46,51,62]. A meta-analysis that included more than 3000 COVID-19 patients showed a pooled mean level of LDH 1.54 times higher for severe illness compared to mild severity [63]. Similarly, our predicted LDH mean level at day 7 of follow-up was approximately 2 times higher in patients who died (478.6 U/L; 95% CI: 397.5, 559.9 U/L) than survivors (259.2; 95% CI: 244.5, 274.0 UI/l). Additionally, elevated baseline LDH levels were associated with higher mortality risk with an HR of 1.30 (95% CI: 1.11, 1.52) for an increase of 100 U/L [64], which is comparable to our estimate (HR from multivariable JM = 1.31; 95% CI: 1.12–1.55 for an increase of 100 U/L).

In summary, decreased lymphocyte count, increased neutrophil count, C-reactive protein, LDH, D-dimer, ferritin, and blood glucose levels were shown to be associated with the mortality of the disease. It has been well assessed that a high level of inflammation is characteristic of COVID-19 pneumonia and the observed biomarkers trend might be the manifestation of this inflammatory response and the subsequent cell damage [65]. In particular, significantly lower lymphocytes and higher neutrophils counts have been widely observed in patients with severe COVID-19 disease in comparison to those who suffered mildly: the lymphopenia might be caused both by the inflammatory mediators, which directly damage the immune system cells and by the migration of the circulating lymphocytes into inflammatory lung tissues [66,67]. Along with these, persistent stimulation by SARS-CoV-2 might lead to lymphocyte exhaustion. High levels of ferritin might reflect macrophage activation, since the synthesis of ferritin is responsive to alterations in cytokine status [68], whereas the increase in LDH in serum is the manifestation of cell necrosis, strongly increased in severe pneumonia [69]. In addition, a recent cohort study conducted in the US on vaccinated and unvaccinated patients infected with SARS-CoV-2 reported significantly lower geometric mean concentrations of several inflammatory biomarkers among the fully vaccinated group than among unvaccinated patients [70]. Likewise, vaccinated patients showed significantly lower levels of ferritin, C-reactive protein, and D-dimer, as well as higher levels of lymphocyte counts than unvaccinated patients [71,72]. It could be interesting to extend the present analysis to vaccinated patients.

The traditional time-to-event analysis using Cox regression can be extended to encompass time-varying covariates (i.e., covariates that are repeatedly measured over the follow-up and their values can change over time), as long as time-varying covariates are exogenous. In the presence of endogenous time-varying covariates, however, the time-dependent Cox regression model could lead to an overestimated effect size (i.e., inflate HRs). The main features of endogenous time-varying covariates are that (i) their existence (and/or future measurements) is directly related to the occurrence (or non-occurrence) of the event of interest, and (ii) they are measured intermittently (i.e., incomplete information occur at random points during the follow-up because, for instance, individuals may skip schedule visits and dropout from the study) [28]. All biomarkers considered in the present study were typical examples of endogenous time-varying covariates. Thus, the JM framework for the simultaneous analysis of the survival data of the event and the longitudinal data of the time-varying covariates is the candidate tool.

The lack of information on patients’ treatment should be counted as a limitation of the present study. Different treatments may have differently modified biomarker levels and consequently their association with the risk of COVID-19 mortality. However, a standard treatment protocol for COVID-19 had not yet been implemented at the time of data collection, maximising the variability due to different drug administration regimes. The general conditions of patients on hospital admission, as measured, for example, by indexes already available in the clinical records or derived by data reported therein should also be included as a study limitation. In addition, information on other baseline characteristics, such as body mass index, smoking habits, and pre-existing comorbidities, were not available. The inclusion of these variables in the model could modify the HR estimates introducing further limits of the present analysis. A limited sample size may have a role in the precision of parameter estimates, especially 95% confidence intervals of HRs for age reported in Table 1.

The main strength of the present work is the use of a multivariable JM to investigate the association between several time-varying covariates (i.e., the set of biomarkers considered), mutually adjusted, and a time-to-event outcome (i.e., COVID-19 mortality).

## 5. Conclusions

Increasing levels of some biomarkers, i.e., neutrophils, C-reactive protein, glucose, and LDH, were significantly associated with higher COVID-19 mortality. Men were at a higher risk of dying than women and age showed the strongest association with a rapid increase in COVID-19 mortality risk from 60 years. These findings using the JM approach confirm the usefulness of biomarkers in assessing COVID-19 severity and mortality. Consequently, prognosis definition and therapy can benefit from the trend pattern analysis of such biomarkers.

## Figures and Tables

**Figure 1 life-14-00343-f001:**
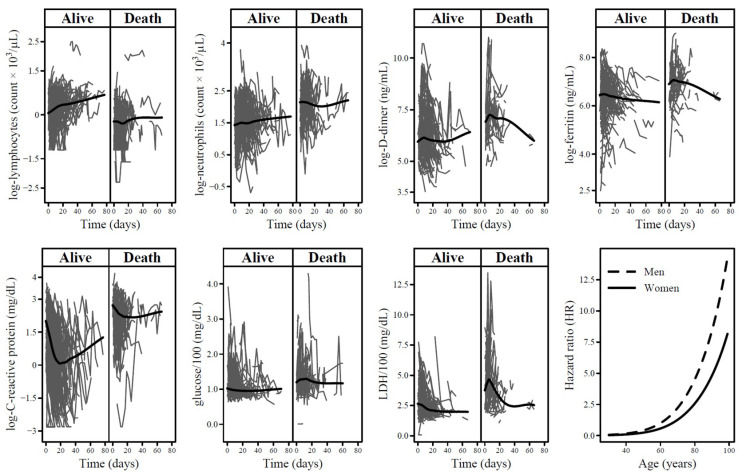
Patient-specific trajectories of each biomarker among 403 COVID-19 patients according to event status (alive/death). The black lines represent the loess smoother. The last panel reports hazard ratios of the association between age and COVID-19 morality in strata of sex. Hazard ratios were estimated by means of the multivariable joint model (JM) with 60 years for men as reference.

**Table 1 life-14-00343-t001:** Distribution of 403 COVID-19 patients: hazard ratio (HR) and corresponding 95% confidence interval (CI) according to age and sex.

Age (Years)	Men (n = 235; 58.3%)				Women (n = 168; 41.7%)			
	At Risk	Mortality Rate ^a^		HR (95% CI) ^b^	At Risk	Mortality Rate ^a^		HR (95% CI) ^b^
<40	14	0.00	3.18 ^c^	Ref	6	0.00	2.27 ^c^	Ref
40–49	26	0.00			6	0.00		
50–59	29	7.59			17	3.32		
60–69	41	10.85		3.55 (1.07, 11.81)	15	4.37		1.84 (0.11, 29.41)
70–79	50	26.72		8.34 (2.86, 24.32)	26	28.44		12.12 (1.58, 93.30)
80–89	62	30.55		9.27 (3.27, 26.32)	71	23.67		10.29 (1.40, 75.49)
≥90	13	88.71		26.88 (8.52, 84.81)	27	43.86		18.74 (2.51, 139.91)

^a^ Calculated per 1000 person-days; ^b^ Estimated using time-invariant Cox regression models separately for men and women including age (in categories) and considering <60 years as reference; ^c^ Mortality rate for age < 60 years.

**Table 2 life-14-00343-t002:** Univariable and multivariable joint models ^a^ (JMs) estimates.

Variables	Univariable JM		Multivariable JM	
	Estimate (95% CI)	*P*	Estimate (95% CI)	*P*
Hematological biomarkers				
Longitudinal submodel: log-lymphocytes (count × 10^3^/µL)				
Intercept	0.68 (0.29, 1.05)	*P* < 0.01	0.64 (0.38, 0.90)	*P* < 0.01
ns(time at measurement in days) 1	1.11 (0.95, 1.27)	*P* < 0.01	1.23 (0.98, 1.49)	*P* < 0.01
ns(time at measurement in days) 2	0.52 (0.40, 0.65)	*P* < 0.01	0.93 (0.47, 1.43)	*P* < 0.01
ns(age in years) 1	−1.47 (−2.20, −0.72)	*P* < 0.01	−1.32 (−1.81, −0.82)	*P* < 0.01
ns(age in years) 2	−0.69 (−0.97, −0.39)	*P* < 0.01	−0.53 (−0.74, −0.30)	*P* < 0.01
Sex (Ref: women)	−0.11 (−0.25, 0.03)	*P* = 0.14	−0.13 (−0.23, −0.04)	*P* < 0.01
Longitudinal sub-model: log-neutrophils (count × 10^3^/µL)				
Intercept	0.90 (0.57, 1.22)	*P* < 0.01	1.09 (0.79, 1.41)	*P* < 0.01
ns(time at measurement in days) 1	−1.10 (−1.40, −0.78)	*P* < 0.01	−0.81 (−1.21, 0.41)	*P* < 0.01
ns(time at measurement in days) 2	−2.89 (−3.55, −2.15)	*P* < 0.01	−2.15 (−2.98, −1.36)	*P* < 0.01
ns(age in years) 1	1.24 (0.65, 1.85)	*P* < 0.01	0.94 (0.32, 1.54)	*P* < 0.01
ns(age in years) 2	0.89 (0.64, 1.15)	*P* < 0.01	0.62 (0.35, 0.89)	*P* < 0.01
Sex (Ref: women)	0.25 (0.13, 0.37)	*P* < 0.01	0.23 (0.12, 0.34)	*P* < 0.01
Coagulation biomarkers				
Longitudinal sub-model: log-D-dimer (ng/mL)				
Intercept	4.34 (3.62, 5.01)	*P* < 0.01	Excluded due to the high number of missing values	
ns(time at measurement in days) 1	−1.72 (−2.02, −1.39)	*P* < 0.01		
ns(time at measurement in days) 2	−2.99 (−3.47, −2.57)	*P* < 0.01		
ns(age in years) 1	4.18 (2.93, 5.52)	*P* < 0.01		
ns(age in years) 2	1.57 (1.01, 2.09)	*P* < 0.01		
Sex (Ref: women)	0.35 (0.10, 0.61)	*P* = 0.01		
Biochemical biomarkers				
Longitudinal sub-model: log-ferritin (ng/mL)				
Intercept	5.18 (4.67, 5.67)	*P* < 0.01	5.27 (4.74, 5.82)	*P* < 0.01
ns(time at measurement in days) 1	−1.05 (−1.32, −0.77)	*P* < 0.01	−1.16 (−1.66, −0.71)	*P* < 0.01
ns(time at measurement in days) 2	−1.69 (−2.23, −1.08)	*P* < 0.01	−2.07 (−3.25, −1.05)	*P* = 0.01
ns(age in years) 1	2.13 (1.15, 3.06)	*P* < 0.01	1.93 (0.82, 2.97)	*P* < 0.01
ns(age in years) 2	0.20 (−0.18, 0.59)	*P* = 0.31	0.16 (−0.29, 0.63)	*P* = 0.52
Sex (Ref: women)	0.53 (0.34, 0.70)	*P* < 0.01	0.56 (0.35, 0.76)	*P* < 0.01
Longitudinal sub-model: log-C-reactive protein (mg/L)				
Intercept	−0.19 (−0.71, 0.34)	*P* = 0.47	0.29 (−0.32, 0.91)	*P* = 0.36
ns(time at measurement in days) 1	−4.23 (−5.09, −3.35)	*P* < 0.01	−5.16 (−6.47, −3.90)	*P* < 0.01
ns(time at measurement in days) 2	1.03 (−0.65, 2.64)	*P* = 0.23	−0.84 (−3.67, 1.79)	*P* = 0.51
ns(age in years) 1	4.31 (3.30, 5.32)	*P* < 0.01	3.67 (2.51, 4.83)	*P* < 0.01
ns(age in years) 2	0.98 (0.57, 1.40)	*P* < 0.01	0.70 (0.20, 1.24)	*P* < 0.04
Sex (Ref: women)	0.50 (0.31, 0.68)	*P* < 0.01	0.38 (0.13, 0.61)	*P* < 0.01
Longitudinal sub-model: glucose/100 (mg/dL)				
Intercept	0.94 (0.45, 1.44)	*P* < 0.01	0.99 (0.75, 1.25)	*P* < 0.01
ns(time at measurement in days) 1	−0.33 (−0.45, −0.20)	*P* < 0.01	−0.40 (−0.64, −0.17)	*P* < 0.01
ns(time at measurement in days) 2	−0.35 (−0.52, −0.20)	*P* < 0.01	−0.61 (−1.23, −0.04)	*P* = 0.04
ns(age in years) 1	0.57 (−0.42, 1.49)	*P* = 0.25	0.41 (−0.08, 0.88)	*P* = 0.11
ns(age in years) 2	0.08 (−0.30, 0.46)	*P* = 0.66	0.05 (−0.15, 0.24)	*P* = 0.64
Sex (Ref: women)	0.02 (−0.16, 0.21)	*P* = 0.80	0.04 (−0.04, 0.14)	*P* = 0.33
Longitudinal sub-model: LDH/100 (U/L)				
Intercept	2.09 (1.56, 2.63)	*P* < 0.01	2.65 (1.76, 3.49)	*P* < 0.01
ns(time at measurement in days) 1	−2.45 (−2.70, −2.19)	*P* < 0.01	−2.45 (−3.21, 1.69)	*P* < 0.01
ns(time at measurement in days) 2	−3.74 (−4.77, −2.80)	*P* < 0.01	−3.32 (−5.36, −1.35)	*P* < 0.01
ns(age in years) 1	2.39 (1.36, 3.40)	*P* < 0.01	1.16 (−0.47, 2.91)	*P* = 0.16
ns(age in years) 2	0.86 (0.46, 1.25)	*P* < 0.01	0.44 (−0.33, 1.21)	*P* = 0.23
Sex (Ref: women)	0.27 (0.07, 0.47)	*P* = 0.01	0.29 (−0.01, 0.61)	*P* = 0.05
**Variables**	**log-hazard ratio (95% CI)**	** *P* **	**log-hazard ratio (95% CI)**	** *P* **
Survival sub-model				
Baseline characteristics				
ns(age in years) 1	-	-	9.12 (2.17, 16.30)	*P* < 0.01
ns(age in years) 2	-	-	3.96 (2.53, 5.38)	*P* < 0.01
Sex (Ref: women)	-	-	0.56 (0.07, 1.03)	*P* = 0.03
Hematological biomarkers		-		
log-lymphocytes (count × 10^3^/µL)	−0.78 (−1.11, −0.44)	*P* < 0.01	0.02 (−0.43, 0.52)	*P* = 0.96
log-neutrophils (count × 10^3^/µL)	1.56 (1.21, 1.91)	*P* < 0.01	0.83 (0.18, 1.52)	*P* = 0.02
Coagulation biomarkers				
log-D-dimer (ng/mL)	0.48 (0.31, 0.64)	*P* < 0.01	Excluded due to the high number of missing values	
Biochemical biomarkers				
log-ferritin (ng/mL)	0.55 (0.33, 0.79)	*P* < 0.01	−0.12 (−0.45, 0.22)	*P* = 0.46
log-C-reactive protein (mg/L)	1.34 (1.05, 1.67)	*P* < 0.01	0.58 (0.25, 0.96)	*P* < 0.01
glucose/100 (mg/dL)	1.15 (0.78, 1.51)	*P* < 0.01	0.98 (0.37, 1.60)	*P* < 0.01
LDH/100 (U/L)	0.59 (0.49, 0.69)	*P* < 0.01	0.27 (0.11, 0.44)	*P* < 0.01

^a^ Univariable and multivariable JMs included age modelled using a natural cubic spline (ns) with 2 knots, and sex (baseline characteristics) and biomarkers (time-varying covariates). In the longitudinal sub-models, time at measurement of biomarkers was modelled using a natural cubic spline (ns) with 2 knots. The log-hazard ratio estimates of the association between biomarkers and COVID-19 mortality are reported in the survival sub-model section. The log-hazard ratio estimates for age and sex of the univariate models are not reported.

## Data Availability

The data presented in this study are available on motivated request from the corresponding author. The data are not publicly available due to privacy restrictions.

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
