# Peer review of "Assessing the Association between Biomarkers and COVID-19 Mortality Using the Joint Modelling Approach"

_life, 2024, doi:10.3390/life14030343_

Round 1

Reviewer 1 Report

Comments and Suggestions for Authors

The results that increasing levels of some biomarkers were significantly associated with higher COVID-19 mortality while pointing out that men were at higher risk of dying than women and age had the strongest association with a rapid increase of COVID-19 mortality risk from 60 years appear to be soundly obtained through suitable statistical techniques.

Some points, however, should be elaborated or explained more clearly and in more detail. In particular:

- The very high value and extremely wide confidence intervals of HRs for older age groups (although the overestimation of effects using standard time-dependent Cox models is briefly addressed in the Discussion) should be justified.

- There is confusion between the legend in Figure 1, eighth panel, and the actual reported measure (HRs vs. RRs); also, it is unclear how the reference value is considered. The same happens with the comment at the end of the Results section.

- In the results for the univariate survivorship models, no indication is given even in the text to the significance and effect of patients' age and sex.

- Limitations of the study presented at the end of the "Discussion" Section should also include the fact that one could include among possible confounders the general condition of patients on admission, as measured, for example, by indices already available in the clinical records or assessable by data reported therein.

In general, more emphasis should be placed on the fact that the study does not introduce any significant novelty regarding the role of biomarkers as risk factors for the course of COVID-19, but shows the use of a statistical methodology best suited for the purpose.

Comments on the Quality of English Language

English requires quick revision as the text has some misprints and some sentences are not expressed clearly and fluently.

The paragraph from lines 253 to 266 should be rewritten in a more orderly and consequential manner.

Reviewer 2 Report

Comments and Suggestions for Authors

After careful consideration, I fell that the manuscript entitled “Biomarkers for predicting COVID-19 mortality using the joint modelling approach” has merit, but presents some problems and is not suitable for publication as it currently stands. Therefore, my decision is "Major Revision."

Here are my comments:

- Statistical analysis section:

     * In the time-dependent Cox regression, which covariates are time-dependent? Describe these variables in the text.

     * The authors should describe under what conditions the result is considered significant. When p< 0.05? Or when the CI does not contain 0 or 1? What level of significance considered? And in the case of the Bayesian model? What was the definition of a significant association? Clarify these points in the manuscript.

- Table 1. It is not clear whether the analysis considered a Cox regression or time-dependent Cox model. Please describe more clearly.

 - Table 2:

     * In Bayesian inference, the intervals should be credibility intervals and not confidence intervals. Correct the entire manuscript.

     * Which credibility interval was considered in bayesian analysis (the author did not describe which type of interval was considered: symmetric CI or HPD - highest posterior density)?

     * Furthermore, there is no p-value in Bayesian statistics. Please review the techniques used in the analyses.

     * What are "Effect" values? What do these values represent? Authors should describe them in the text of the manuscript.

     * point out the reference levels of categorical variables in Table 2. For example, there is no way of knowing whether the reference category for the "Sex" variable is male or female.

     * In the survival sub-model, what does "log-hazard" mean? Ideally, the authors should not present the values on the log scale. In fact, because they are on the logarithmic scale, it is difficult for non-specialist readers to understand the similarity between the results in Table 2 and the results described in lines 167 to 181 of the manuscript.

- Figure 1.

     * The last panel plot has the Y axis described as "relative risk (RR)" instead of hazard ratio (HR). Please correct.

     * The footnotes "a" and "b" in the title are not described in the figure. Furthermore, there are two different descriptions for footnotes "a" and "b".

Reviewer 3 Report

Comments and Suggestions for Authors

Maso et al studied the biomarkers for COVID-19 mortality. They analyzed 403 patients with univariate and multivariate biomarkers and observed that all univariate biomarkers are significantly associated with COVID-19 mortalities. In multivariate analysis, some of the biomarkers are associated with mortalities. The data is interesting and the approach to define COVID-19 mortality prediction is acceptable.. However, the following points should be considered with sufficient explanation

1.       In table 1, the HR are increased in woman in 70-79 and 80-89 age group than Man of the same age group. What is the explanation for these anomalies?

2.       In Figure 1, last panel, it seems the RR (relative Risk are almost similar until the age 60 for the woman and Men. What would be the reason for this when men are always at more risk than women?

3.       Author should mention that the effect of medications (anti-viral agent and antibodies)  and physical intervention did not have effect on biochemical factors and other blood cellular counts.

4.       This is the study of patients of early 2020. Did the author used to test/predict mortalities by their model in patients after this period? If yes, they should mention as a validation of their model further patients,

5.        What happens after vaccination? Do neutrophil, hCRP, glucose level still shows higher association after vaccinated patients. Please note that vaccinated patient have  mostly milder COVID-19 phenotype. Author should discuss this in discussion.

6. Moderate English editing is required

Comments on the Quality of English Language

Moderate English editing required

Round 2

Reviewer 2 Report

Comments and Suggestions for Authors

I believe that the authors have made the corrections and/or justified the answers satisfactorily and, therefore, I feel that the manuscript is now suitable for publication in Life.